# Development of High-Quality Nuclei Isolation to Study Plant Root–Microbe Interaction for Single-Nuclei Transcriptomic Sequencing in Soybean

**DOI:** 10.3390/plants12132466

**Published:** 2023-06-28

**Authors:** Leonidas W. D’Agostino, Lenin Yong-Villalobos, Luis Herrera-Estrella, Gunvant B. Patil

**Affiliations:** 1Institute of Genomics for Crop Abiotic Stress Tolerance, Department of Plant and Soil Science, Texas Tech University, Lubbock, TX 79409, USA; ldagosti@ttu.edu (L.W.D.); lenin.yong@ttu.edu (L.Y.-V.); 2Unidad de Genomica Avanzada, Centro de Investigacion y de Estudios Avanzados, Instituto Politecnico Nacional, Irapuato 36821, Mexico

**Keywords:** soybean, nodule, single cell, sNucRNA, nuclei, 10×

## Abstract

Single-nucleus RNA sequencing (sNucRNA-seq) is an emerging technology that has been rapidly adopted and demonstrated to be a powerful tool for detailed characterization of each cell- and sub cell-types in complex tissues of higher eukaryotes. sNucRNA-seq has also been used to dissect cell-type-specific transcriptional responses to environmental or developmental signals. In plants, this technology is being utilized to identify cell-type-specific trajectories for the study of several tissue types and important traits, including the single-cell dissection of the genetic determinants regulating plant–microbe interactions. The isolation of high-quality nuclei is one of the prerequisite steps to obtain high-quality sNucRNA-seq results. Although nuclei isolation from several plant tissues is well established, this process is highly troublesome when plant tissues are associated with beneficial or pathogenic microbes. For example, root tissues colonized with rhizobium bacteria (nodules), leaf tissue infected with bacterial or fungal pathogens, or roots infected with nematodes pose critical challenges to the isolation of high-quality nuclei and use for downstream application. Therefore, isolation of microbe-free, high-quality nuclei from plant tissues are necessary to avoid clogging or interference with the microfluidic channel (e.g., 10× Genomics) or particle-templated emulsion that are used in sNucRNA-seq platforms. Here, we developed a simple, effective, and efficient method to isolate high-quality nuclei from soybean roots and root nodules, followed by washing out bacterial contamination. This protocol has been designed to be easily implemented into any lab environment, and it can also be scaled up for use with multiple samples and applicable to a variety of samples with the presence of microbes. We validated this protocol by successfully generating a barcoded library using the 10× Genomics microfluidic platform from tissue subjected to this procedure. This workflow was developed to provide an accessible alternative to instrument-based approaches (e.g., fluorescent cell sorting) and will expand the ability of researchers to perform experiments such as sNucRNA-seq and sNucATAC-seq on inherently heterogeneous plant tissue samples.

## 1. Introduction

Soybean (*Glycine max* L.) is an important crop that plays a significant role in global food and feed production and the agricultural industry [1]. One of the key factors contributing to the high productivity of soybean plants is their ability to form a symbiotic relationship with nitrogen-fixing bacteria known as rhizobia. This sustainable association allows soybean plants to acquire nitrogen from the atmosphere, which is essential for their growth and development. The process of nitrogen fixation occurs within specialized structures called nodules that form on the plant roots and these tiny allies possess the unique ability to convert atmospheric nitrogen into a form that plants can utilize. The symbiotic relationship between soybean plants and rhizobia begins when the plant roots release specific compounds known as flavonoids into the soil which act as signals, attracting compatible rhizobia to the root system. The rhizobia, in response to these signals, produce signaling molecules called Nod factors, which further stimulate the plant root cells to initiate nodule formation [2].

Despite these amazing mutualistic benefits, the complexity of signal reception, and nodule development, nitrogen uptake at the sub-cellular level is elusive in soybean and other leguminous plants. To gain a deeper understanding of the molecular mechanisms involved in the legume–rhizobium symbiosis and nitrogen fixation, researchers have utilized advanced technologies such as RNA sequencing in several leguminous plants including *Medicago truncatula*, soybean, and *Lotus japonica* and identified a suite of genes that are activated or repressed during nodule development and nitrogen fixation (Zhu et al., 2013; Cabeza et al., 2014; Zhu et al., Yuan et al., 2017; Shi et al., 2020). While bulk RNA sequencing provides valuable insights into the overall gene expression patterns in a tissue or organ, it lacks the resolution to uncover cell-specific gene expression, and this limitation can be addressed by employing single-cell RNA sequencing (scRNA-seq) techniques [3].

Single-cell or single-nuclei RNA sequencing is a powerful tool to study cellular heterogeneity in a multitude of different tissue types. Single-nuclei sequencing allows the analysis of gene expression profiles at the level of individual cells, providing a more detailed understanding of cellular heterogeneity. Therefore, by examining the gene expression profiles of different nodule cells, researchers can identify cell types, unravel specialized functions, and gain insights into the communication and interactions between soybean and rhizobia at a cellular level. Performing single-nuclei sequencing on soybean nodules is crucial because it could enable the identification of rare cell types that might play significant roles in the symbiotic process but are present in small numbers. Secondly, it would allow the characterization of gene expression dynamics in different cell types during nodule development and nitrogen fixation, providing insights into the temporal regulation of symbiotic processes. Most importantly, single-nuclei sequencing facilitates the discovery of novel genes and regulatory networks, which may contribute to the development of improved symbiotic associations and enhanced nitrogen fixation efficiency in soybean and other leguminous crops. Recently, it has become increasingly utilized in different plant species to characterize gene regulation at the single-nuclei level. Species such as *Arabidopsis thaliana* [4,5,6,7,8,9,10], *Zea mays* [11,12,13], *Oryza sativa* [14,15,16], and *Nicotiana attenuata* [17] make up the majority of data generated to date when it comes to plant tissue studied.

The prerequisite for generating single-nuclei transcriptome data is high-quality cells. More recently, the transition from using whole cells, i.e., protoplasts, to using nuclei has been observed. Using protoplasts for single-cell sequencing can have certain disadvantages, for instance, the isolation depends upon the enzymatic digestion of cell walls that can lead to altered gene expression due to cellular stress, and there is an increased risk of protoplasts bursting in the process [18]. The use of nuclei bypasses these potential pitfalls and provides a better snapshot of unaltered gene expression at the point of isolation, which can be more directly correlated with ATAC-Seq data [19]. Isolation of nuclei from different species often requires optimized protocols based on the cellular contents of the target organism. This process is inherently more complicated in plants than in animal tissues due to the presence of plant-specific organelles and components such as the chloroplasts and cell wall. Moreover, it becomes increasingly difficult to obtain high-quality nuclei when plant tissue is associated with beneficial or pathogenic microorganisms [20]. For example, root tissues colonized with rhizobium bacteria (nodules), leaf tissue infected with bacterial or fungal pathogens, or roots infected with nematodes, bacteria, or fungi possess critical challenges for the isolation of high-quality nuclei for downstream applications, including microfluidic-based sNucRNA-seq. Therefore, isolation of high-quality nuclei from plant tissues free of microbial cells is necessary to avoid incorrect nuclei counting, interference of microorganism particles (cell debris), and prevention of clogging of the microfluidic channel of droplet-based sNucRNA-seq platform (e.g., 10× Genomics).

Conventionally, researchers use cell sorting technology to separate the cells/nuclei from other cellular debris or microbes. As reviewed by Chen et al. [21], currently, several different methods are available for sorting single cell/nuclei with limiting dilution, micromanipulation using Fluorescent-Activated Cell Sorting (FACS), Fluorescent-Activated Nuclei Sorting (FANS), and laser capture microdissection. However, each method has its advantages and disadvantages [21]. The most widely used workaround implemented by researchers is the previously mentioned FACS or FANS, and these methods have been used for sorting nuclei within tissue samples for several plant, human, and animal tissue types [3,22,23,24]. While FANS is a relatively effective method for separating nuclei from other cellular debris, and in this case bacteria, there are several drawbacks to this technology that include: (1) The cost associated with the instrument and reagents; (2) Additional steps for staining and time required for sorting; and (3) The yield rate for sorted cells/nuclei. Moreover, these factors may significantly affect the final yield and sNucRNA library preparation, data analysis, and recovery of false positives. Furthermore, in most instances the recovery rate of viable plant cells is low and variable, so a significantly larger number of cells are needed when compared to non-FACS nuclei isolation. Hence, fluorescent sorting methods require large amounts of starting tissue, which can be a significant bottleneck for specific studies where the starting material is limited [21,25]. In addition to this, the loss of significant amounts of nuclei can potentially lead to the exclusion of rare cell types in a sample leading to false positive gene-network predictions. Likewise, the laser capture microdissection has similar disadvantages. These processes are often costly where the quality fluorescent cell sorters cost anywhere from $100,000 to $350,000 and the cost per sample range from $50–$80, and in comparison, the only additional costs of this protocol include percoll. Additionally, cell-sorting method is relatively time consuming and not amenable to high-throughput settings. Furthermore, skilled technicians are needed to use and maintain the instruments, which is another roadblock to their widespread use.

To understand and dissect the cell-specific dynamics of gene expression in soybean–rhizobium association, we optimized first-in-class method for the isolation of high-quality nuclei from soybean root nodule tissue without the need for any uncommon instrumentation and additional processing. This process excludes the prohibitive requirements for fluorescent sorting, such as high amounts of starting material or lengthy purification steps. The performance of this method was evaluated by nuclei yield via cell counting and RNA integrity via Tapestation QC from cDNA. Overall, this process can be completed in less than one hour and requires only easily obtained reagents, making previously restrictive experiments more accessible.

## 2. Experimental Design

The soybean cultivar “Williams 82” was used for all experimental procedures. The Williams 82 is the gold-standard reference genome for soybean. Seeds were propagated in the Phytotron greenhouse facility at Texas Tech University under 16/8 h day/night conditions. Soybeans seeds were planted in sterile potting mix to facilitate germination. Five days after emergence, soybean plants were inoculated with rhizobium strain “*Bradyrhizobium diazoefficiens* USDA 110” via root dip method, as detailed in Li et al. [26]. Briefly, the plantlets were removed from the soil, and the roots submerged in rhizobium culture for 90 min in OD at _600_ = 0.8 concentration of *B. diazoefficiens* USDA 110 resuspended in nitrogen-free Hoagland media. Plantlets were then transplanted into two-gallon pots with sterile sand and supplemented with a nitrogen-free fertilizer. Nodules were collected at several intervals, including at 2-, 4-, 6-, and 8-weeks post-inoculation (wpi), and the nodule as well as the corresponding root tissue were used for nuclei isolation.

## 3. Materials and Equipment

### 3.1. Chemical and Stock Solutions

PIPES (0.5 M, pH 6.0): Sigma-Aldrich Cat. No. P6757. For 50 mL, dissolve 7.56 g of 1,4-Piperazinediethanesulfonic acid in 30 mL sdH_2_O and adjust the volume to 50 mL. Adjust pH to 6.0 with Potassium Hydroxide (KOH) (see Note 1,2).EGTA (0.25 M, pH 6.0): Sigma-Aldrich Cat. No. 4100. For 50 mL, dissolve 4.76 g of Ethylene glycol bis(2-aminoethyl ether)-N,N,N′,N′-tetraacetic acid in 30 mL sdH_2_O and adjust the volume to 50 mL. Adjust pH to 6.0 with Hydrogen Chloride (HCl) (see Note 1,3).DTT (0.5 M): Sigma-Aldrich Cat. No. 3860. Dissolve 771.25 mg dithiothreitol in 8 mL of sdH_2_O and adjust the volume to 10 mL. Dispense into 1 mL aliquots and store in the dark at −20 °C (see Note 4).Hexylene Glycol: Sigma-Aldrich Cat. No. 68340.L-Lysine (1 M): Sigma-Aldrich Cat. No. 62840. For 50 mL, dissolve 7.31 g of L-Lysine in 30 mL sdH_2_O and adjust the volume to 50 mL (see Note 1,2).Liquid NitrogenMACS BSA Stock Solution: Miltenyi Biotec Cat. No. 130-091-376. (see Note 4)Magnesium Chloride—Hexahydrate: Sigma-Aldrich Cat. No. M0250. For 50 mL, dissolve 5.08 g in 30 mL sdH_2_O and adjust the volume to 50 mL (see Note 1,3).Propidium Iodide: Sigma-Aldrich Cat. No. 537059. Dissolve 5 mg in 4 mL sdH_2_O and adjust the volume to 5 mL. Dispense into 20 μL aliquots and store in the dark at −20 °C.Nitrogen-free Hoagland media: BIO WORLD Cat. No. 30630038-5Percoll: Sigma-Aldrich Cat. No. P1644.PBS 1× Phosphate-Buffered Saline: Corning Cellgro Cat No. 21-040-CV.Protector RNAse inhibitor: Sigma-Aldrich Cat. No. 3335402001.Sodium Metabisulfite: Sigma-Aldrich Cat. No. S9000.Sodium Diethyldithiocarbamate—trihydrate: Sigma-Aldrich Cat. No. 228680.

### 3.2. Solutions and Media

Prepare working solutions as shown in Table 1.

### 3.3. Other Supplies and Equipment

50 mL conical centrifuge tubes [Thermo-scientific].60 mm disposable Petri dishes [Fisherbrand].Cell strainer 10 μm [Pluriselect].Connector ring [Pluriselect].Fluorescent microscope with 40× lens [e.g., Invitrogen EVOS M5000].MACS SmartStrainers (30 μm, 70 μm, 100 μm) [Miltenyi Biotec].Microcentrifuge [e.g., Eppendorf 5424 R].Miracloth [Sigma-Aldrich].Mortar and pestle.Platform shaker [ex. Innova 2000].Premium 2.0 mL MCT graduated natural microcentrifuge tubes [Fisherbrand].Regular duty single edge razor blades.Regular pipette tips, 1000 μL, 200 μL, 10 μL.SKC, Inc. C-Chip™ Disposable Hemacytometers [Fisher-scientific].Small paintbrush.Swinging rotor centrifuge [e.g., Eppendorf 5910 R].Syringe.Wide bore pipette tips 1000 μL, 200 μL.

## 4. Detailed Procedure and Results

### 4.1. Nuclei Isolation

#### 4.1.1. Tissue Homogenization and Filtration

Always prepare a fresh Nuclei Isolation Buffer (NIB). The NIB used in this study was modified from Peterson et al. [27]. Similarly, Wash Buffer (WB), 30% percoll NIB, and 10% percoll WB are prepared fresh as described in Table 1. Prepare the other supplies needed as well (*see*
**Notes 5, 6**).For frozen root or nodule tissue, take approx. 500 mg tissue and grind to a fine powder in a mortar and pestle using liquid nitrogenFor fresh root or nodule tissue, place approx. 500 mg tissue in a 60 mm sterile Petri dish and finely chop with a razor blade in 500 μL of ice-cold NIB for 4–5 min (*see*
**Note 7, 8, 9**).For the frozen tissue, transfer the powder to a 50 mL conical tube containing 5 mL of NIB and incubate for 10 min on ice shaking at a gentle RPM. For fresh tissue, add 4.5 mL NIB to the Petri dish and incubate for 5 min on ice shaking at a gentle RPM (*see*
**Note 9, 10**).Slowly decant the lysate through one layer of miracloth followed by a 100 μm, 70 μm, and 30 μm cell strainer (*see*
**Note 11, 12**).With a wide-bore tip, transfer the flowthrough to a new 50 mL tube with a 10 μm cell strainer. Negative pressure can be applied with the connector ring and syringe to aid filtering through the strainer (see **Note 12**).

#### 4.1.2. Nuclei Washing

a. Pellet the filtrate at 1000× *g* for 5 min in a fixed angle microcentrifuge at 4 °C (*see*
**Note 9**).

b. Carefully underlay the filtrate with 5 mL of 30% percoll NIB. Pellet the nuclei at 600× *g* for 10 min. (*see*
**Note 9**).

Discard the supernatant and slowly resuspend the pellet in 1 mL ice cold WB with a wide-bore tip and transfer to a 2 mL microcentrifuge tube (*see*
**Note 9**).Spin at 1000× *g* for 2 min at 4 °C.

a. Gently resuspend root nuclei pellet in 250 μL WB and observe nuclei as indicated in step 3.2. (*see*
**Note 9**).

b. Gently resuspend pellet in 1 mL 10% percoll WB (*see*
**Note 9**).

3.Spin at 1000× *g* for 2 min at 4 °C. Discard the supernatant and resuspend in 1 mL of 10% percoll WB (*see*
**Note 9**).4.Repeat step 5.b. for a total of 3 washes. (*see*
**Note 9, 13**)5.Gently resuspend nodule nuclei pellet in 250 μL WB and proceed with assessment.

#### 4.1.3. Notes and Critical Parameters

These solutions can be stored at 4 °C for one month.Filter sterilize using a 0.22 µm syringe filter. Simple Pure brand filters were used in this study [Product# SFNY025022S].Sterilize via autoclaving on liquid cycle—121 °C for 15 min.Working Stock can be stored at 4 °C for 2–3 weeks.Nuclei Isolation buffer should be prepared fresh and not stored for more than 48 h at 4 °C.All solutions, tubes, mortar and pestle, and cell strainers should be pre-chilled to 4 °C for at least 1 h and kept on cold conditions during the whole process. Centrifuge rotors should be cooled to the same temperature.500 mg of nodule tissue was determined to be the best starting weight for nuclei isolation. When testing, more starting tissue (>600 mg) proved too difficult to wash away enough bacteria. Smaller amounts of tissue (<400 mg) yielded too few nuclei in final suspension. The same amount is used for root tissue as an equal comparison.Do not allow the tissue to thaw during grinding. Keep adding liquid nitrogen after 5–6 rounds of grinding.Steps with (a) connote steps for root tissue. Steps with (b) connote modified or additional steps for nodule tissue.A small paintbrush was used to facilitate the tissue powder into the 50 mL round-bottom tube. It can also be used to push all the tissue into the NIB.A small hole was made in a 50 mL conical tube cap in order to make a holder for the miracloth and used in conjunction with the Miltenyi biotech cell strainers, which are stackable. With non-stackable filters, each size strainer will be placed in individual tubes. See Figure 1.Cell strainers and miracloth should be prewetted with Wash Buffer before filtering. The BSA in the buffer assists with filtration, especially with the 10 μm strainer.Repeating the wash step three times was determined to be the most consistent number of washes to produce high quality and quantity of nuclei suspension. However, the washing step can be extended if needed. When checked under 40× magnification, additional numbers of washes past three showed diminishing amounts of bacteria removed and an increase in nuclei lost and thus is not recommended.

### 4.2. Nuclei Quantity and Quality Assessment

Since nuclei isolation is a standard procedure used in molecular and cellular biology, we initially used standard protocols that use either liquid nitrogen for tissue grinding [28,29] and use fresh tissue chopping [30,31,32]. Following these methods, we recovered enough nuclei, but overwhelming presence of bacteria consistently remained (see Figure 2A). Quantifying the nuclei in this state presented a challenge as stains like DAPI and Propidium Iodide (PI) would stain the nuclei as well as the dead bacteria. DAPI is a DNA stain that is often used in assays to visualize bacterial viability along with PI, which permeates the dead bacteria’s damaged membrane to stain DNA [33,34,35]. Due to this non-specific staining of both nuclei and bacteria, automatic cell counters cannot differentiate between the two, which would necessitate the use of manual cell counting methods. However, as previously mentioned, even if an accurate count of nuclei was obtained from a sample such as Figure 2A, the sample would cause clogging of microfluidic channels, which was observed in earlier experiments. Therefore, the protocol presented here helped removing all (>98%) the bacterial debris from the sample (Figure 2B). Since the nuclei are free from profuse bacterial presence, they can be accurately counted via manual methods or automatic cell counters. Using 500 mg of nodule tissue either prepared fresh or frozen, the final yield was consistently 2.5 million on average in total. We compared the yield loss between the percoll precipitation steps and found that about 5–8% nuclei were lost between the initial 30% percoll NIB wash (nuclei washing step 1b) and the 10% percoll WB washing steps (nuclei washing step 4b–6b). In the initial 30% percoll NIB wash, approx. 60–70% of bacteria are washed off and no more than 5% of total are lost. During the three 10% percoll WB washing steps, each step was observed to remove approx. 10% of the remaining bacteria with a loss of no more than 1.5% per wash step. Yield loss was calculated by counting nuclei and bacteria manually using a Neubauer hemacytometer across >20 replications of different time points and fresh and frozen tissue.

To determine the quality of the isolated nuclei, 10 μL of suspension was mixed with 2 μL of Propidium Iodide and incubated for 5 min on ice. A total of 10 μL of this mixture was then placed on a glass microscope slide with a cover slip. Observation of the stained nuclei was performed under RFP and brightfield at 40× magnification (EVOS M5000, Invitrogen, was used in this study). As shown in Figure 2C good-quality nuclei appear as spherical, white, and intact objects. For quantification of nuclei isolated, 10 μL of suspension is loaded onto an improved Neubauer hemacytometer and manually counted. A dilution of the nuclei suspension is made in WB to meet the single-nuclei RNA sequencing platform expectation suggested concentration and confirmed with an additional hemacytometer count. Specific to this protocol, during the 10× Genomics single-nuclei microfluidics pipeline, a quality control (QC) is generated at the end of library preparation (Tapestation D1000), and a QC is performed to determine library quality and size before sequencing using standard Tapestation. As shown in Figure 3 for QC results, we successfully prepared the single-nuclei library with an average of 458 bp fragments. The library construction QC was generated from nodule tissue and exhibits an anticipated main peak between 300–700 bp as per single-nuclei RNA sequencing expectation.

## 5. Summary

In summary, the symbiotic relationship between soybean plants and rhizobia and the subsequent process of nitrogen fixation within nodules are vital for soybean growth and productivity. RNA sequencing, particularly single-cell sequencing, plays a crucial role in unraveling the complex molecular mechanisms underlying this symbiotic interaction. By employing these advanced techniques, researchers can gain valuable insights into the cellular dynamics, gene expression patterns, and regulatory networks associated with soybean nodules, ultimately leading to the development of strategies for improving nitrogen fixation efficiency and enhancing soybean crop yields. With these aims, we have successfully developed a working protocol to isolate high-quality nuclei from one of the most complex tissues in soybean (nodule) and further performed library preparation suitable for performing single-nuclei RNA sequencing. This method can be applied to a variety of plant tissues that are challenged with various beneficial (for example, rhizobium and other microbes) and pathogenic (fungi, nematodes, and bacteria) microorganisms. Most importantly, researchers do not need to depend on expensive Fluorescent-Activated Nuclei Sorting (FANS) to separate the desired nuclei from microbial cells.

## Figures and Tables

**Figure 1 plants-12-02466-f001:**
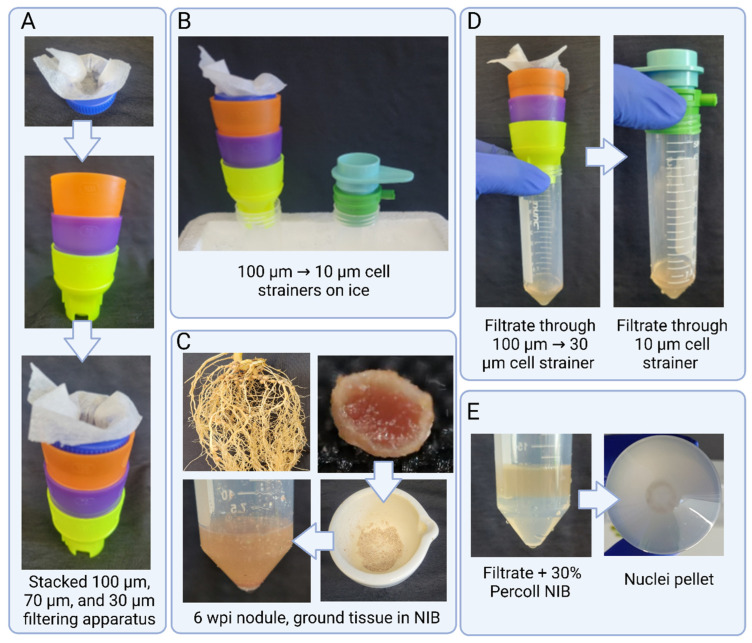
Steps involved in filtration. (**A**) Miracloth in makeshift holder placed in cell strainer stack. (**B**) Cell strainer stack and 10 µm strainer with connecter ring on ice. (**C**) The 6-week post-inoculation (wpi) nodule tissue shown ground to powder and placed in NIB (**D**) Filtrate after passing through cell strainers. (**E**) Filtrate underlaid with 30% percoll NIB and pellet after spinning for 600× *g* for 10 min.

**Figure 2 plants-12-02466-f002:**
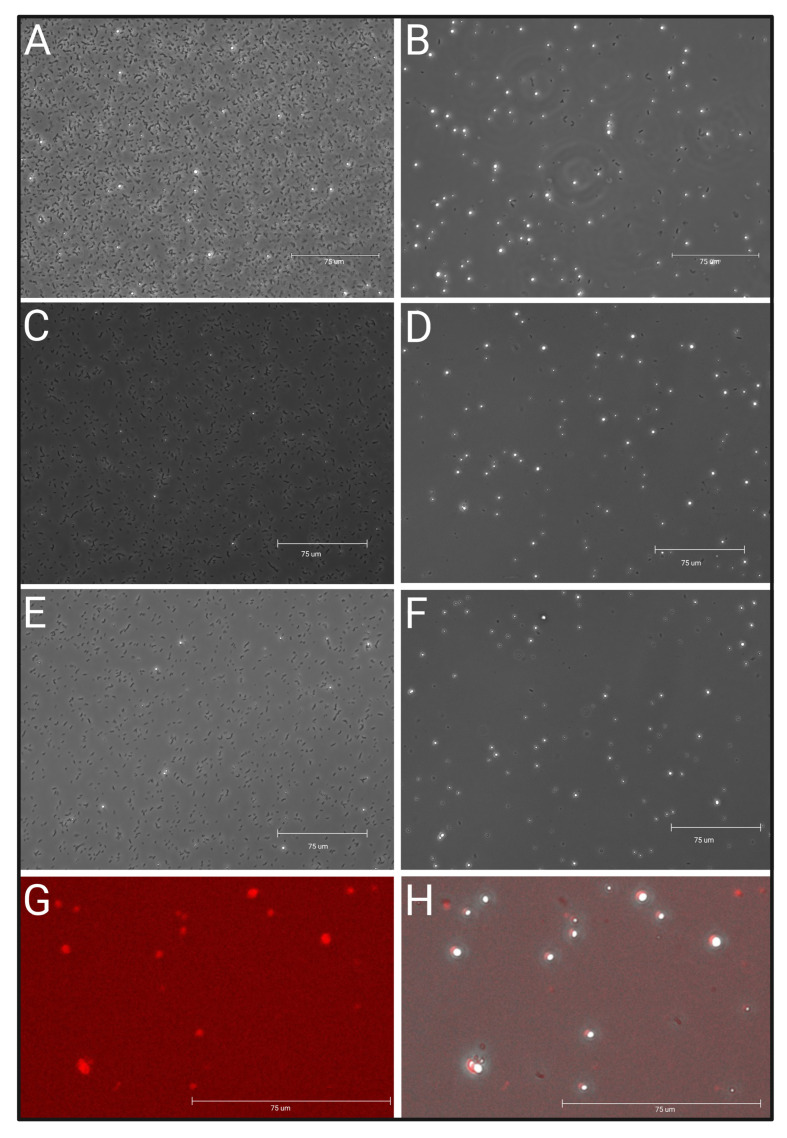
Nuclei observed under the microscope before and after bacterial clean up portion of the protocol. Magnification 40×. (**A**) Nuclei and bacteria suspension from 500 mg 8 wpi fresh nodule tissue in 1 mL WB after tissue chopping and filtration step 6. (**B**) Suspension from (**A**) in 250 µL WB after nuclei washing step 7b. (**C**) Nuclei and bacteria suspension from 500 mg 6 wpi frozen nodule tissue in 1 mL WB after tissue homogenization and filtration step 6. (**D**) Suspension from (**C**) in 250 µL WB after nuclei washing step 7b. (**E**) Nuclei and bacteria suspension from 500 mg 2 wpi frozen nodule tissue in 1 mL WB after tissue homogenization and filtration step 6. (**F**) Suspension from (**E**) in 250 µL WB after nuclei washing step 7b. (**G**) PI-stained nuclei under RFP filter. (**H**) Overlay of C and D shows PI-stained nuclei in red.

**Figure 3 plants-12-02466-f003:**
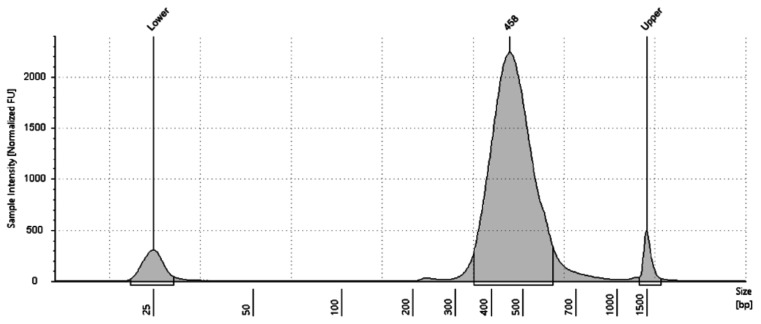
Sample quality control of single-nuclei sequencing library preparation analyzed with a high-sensitivity Tapestation. The graphs display library size distribution after cDNA amplification.

**Table 1 plants-12-02466-t001:** Reagents and solutions for nuclei isolation and purification. *see*
**Notes 1 to 5** for solution storage and autoclaving.

#	Chemical	Stock Concentration	Final Concentration
**A**	**Nuclei Isolation Buffer—pH 7.0**		
1	Sodum Metabisulfite	-	10 mM
2	Sodium Diethyldithiocarbamate	-	0.50%
3	Magnesium Chloride	0.5 M	10 mM
4	EGTA (pH 6.0)	0.25 M	6 mM
5	L-Lysine	1 M	200 mM
6	PIPES (pH 6.0)	0.5 M	10 mM
7	Hexylene glycol	-	1 M
8	DTT	0.5 M	1 mM
9	RNAse Inhibitor	-	0.2 Units/μL
**B**	**Wash Buffer (WB)**		
1	PBS	-	1×
2	BSA	10%	0.10%
3	DTT	0.5 M	1 mM
4	RNAse Inhibitor	-	0.2 Units/μL
**C**	**30% percoll NIB**		
1	NIB	-	70%
2	Percoll	-	30%
**D**	**10% percoll Wash Buffer**		
1	WB	-	90%
2	Percoll	-	10%

## Data Availability

Not applicable.

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
