# Peer review of "Development of High-Quality Nuclei Isolation to Study Plant Root–Microbe Interaction for Single-Nuclei Transcriptomic Sequencing in Soybean"

_plants, 2023, doi:10.3390/plants12132466_

Round 1

Reviewer 1 Report

The authors developed a working protocol to isolate high-quality nuclei from one of the soybean nodules and provided a detailed protocol along with the material list. The authors mentioned this protocol is less expensive than FANS; it would be worth comparing the cost difference between these two protocols. 

Also, there are some typing errors:

1. Figure 1. 30 um filtering apparatus

2. Line 324 space between 500 mg (Nuclei and bacteria suspension from 500mg nodule tissue in 1 mL)

3. Align the texts in the tables to the left.

Author Response

The authors developed a working protocol to isolate high-quality nuclei from one of the soybean nodules and provided a detailed protocol along with the material list. The authors mentioned this protocol is less expensive than FANS; it would be worth comparing the cost difference between these two protocols. 

We thank reviewer’s time to review our manuscript and appreciate the feedback. As suggested we have added the cost range for the instrument and also the cost per sample from the core facilities.

Also, there are some typing errors:

  1. Figure 1. 30umfiltering apparatus

Corrected as suggested by reviewer.

  1. Line 324 space between 500 mg(Nuclei and bacteria suspension from 500mg nodule tissue in 1 mL)

Corrected as suggested by reviewer.

  1. Align the texts in the tables to the left.

Corrected as suggested by reviewer.

Reviewer 2 Report

This manuscript describes a method to isolate soybean root nuclei with less or no contamination of bacteria. As the authors stated in the manuscript, it is essential to generate high-pure and high-quality nuclei for RNA-seq. This manuscript nicely shows a  procedure to remove bacterial contamination from the nuclei isolated from soybean roots and nodules filled with rhizobia.

I have a few comments regarding reproducibility and quality check. The authors mention quantitative values but they are not really supported by figures (all qualitative). The reproducibility of the results (replicates) is not mentioned throughout. I listed a couple that I identified.

1. Lines 271-272: Fig. 2B somehow shows >98% removal of bacteria. The image itself does not show any quantitative data. How is this reproducible?

2. Lines 275-281: I suggest showing these data with replicates. 

Author Response

This manuscript describes a method to isolate soybean root nuclei with less or no contamination of bacteria. As the authors stated in the manuscript, it is essential to generate high-pure and high-quality nuclei for RNA-seq. This manuscript nicely shows a procedure to remove bacterial contamination from the nuclei isolated from soybean roots and nodules filled with rhizobia.

I have a few comments regarding reproducibility and quality check. The authors mention quantitative values but they are not really supported by figures (all qualitative). The reproducibility of the results (replicates) is not mentioned throughout. I listed a couple that I identified.

We thank reviewer’s time to review our manuscript and appreciate the feedback. As suggested we have added we have elaborated on the quantitative measurements. We have tested our method for more than 5 times and during each experiment we use more than 20 manual counts to calculate the nuclei yield.

  1. Lines 271-272: Fig. 2B somehow shows >98% removal of bacteria. The image itself does not show any quantitative data. How is this reproducible?
  2. Lines 275-281: I suggest showing these data with replicates. 

Thank you for your comment, we have added the additional figures to highlight the reproducibility with nodule tissue. Importantly, our method is efficient with different time points of nodule development and with frozen or fresh tissue samples. As per the reviewer’s suggestion, we have added this information to the manuscript.

Reviewer 3 Report

This paper developed a simple, effective, and  efficient method to isolate high-quality nuclei from soybean roots and root nodules. The method can be used elsewhere.  There are many mistakes in the format of literature. Need to revise one by one.

Author Response

This paper developed a simple, effective, and  efficient method to isolate high-quality nuclei from soybean roots and root nodules. The method can be used elsewhere.  There are many mistakes in the format of literature. Need to revise one by one.

We thank the reviewer’s time to review our manuscript and appreciate the positive feedback. We have formatted all the references as per MDPI standard format using EndNote program. Also, the manuscript is reviewed by native English speaking scientist. Again, many thanks for your efforts to review this manuscript.